# Predictive Performance of Mobile Vis–NIR Spectroscopy for Mapping Key Fertility Attributes in Tropical Soils through Local Models Using PLS and ANN

**Mateus Tonini Eitelwein [1], Tiago Rodrigues Tavares [2]**, **José Paulo Molin [3],\***, **Rodrigo Gonçalves Trevisan [1]**, **Rafael Vieira de Sousa [4]** and **José Alexandre Melo Demattê [5]**

1   Smart Agri Technological Solutions, Av. Limeira, 1131, 3rd Office, Piracicaba 13414-018, Brazil; mateus@smart.agr.br (M.T.E.); rodrigo@smart.agr.br (R.G.T.)
2   Laboratory of Nuclear Instrumentation (LIN), Center for Nuclear Energy in Agriculture (CENA), University of São Paulo (USP), Piracicaba 13416-000, Brazil; tiagosrt@usp.br
3   Laboratory of Precision Agriculture (LAP), Department of Biosystems Engineering, Luiz de Queiroz College of Agriculture (ESALQ), University of São Paulo (USP), Piracicaba 13418-900, Brazil
4   Robotics and Automation Group, Department of Biosystems Engineering, Faculty of Animal Science and Food Engineering, University of São Paulo (USP), Pirassununga 13635-900, Brazil; rafael.sousa@usp.br
5   Geotechnologies in Soil Science Group (GeoCis), Soil Science Department, Luiz de Queiroz College of Agriculture (ESALQ), University of São Paulo (USP), Piracicaba 13418-900, Brazil; jamdemat@usp.br
\*   Correspondence: jpmolin@usp.br; Tel.: +55-19-3447-8502

**Abstract:** Mapping soil fertility attributes at fine spatial resolution is crucial for site-specific management in precision agriculture. This paper evaluated the performance of mobile measurements using visible and near-infrared spectroscopy (vis–NIR) to predict and map key fertility attributes in tropical soils through local data modeling with partial least squares regression (PLS) and artificial neural network (ANN). Models were calibrated and tested in a calibration area (18-ha) with high spatial variability of soil attributes and then extrapolated in the entire field (138-ha). The models calibrated with ANN obtained superior performance for all attributes. Although ANN models obtained satisfactory predictions in the calibration area (ratio of performance to interquartile range (RPIQ) $\geq$ 1.7) for clay, organic matter (OM), cation exchange capacity (CEC), base saturation (V), and exchangeable (ex-) Ca, it was not repeated for some of them when extrapolated into the entire field. In conclusion, robust mappings (RPIQ = 2.49) were obtained for clay and OM, indicating that these attributes can be successfully mapped on tropical soils using mobile vis–NIR spectroscopy and local calibrations using ANN. This study highlights the need to implement an independent test to assess reliability and extrapolability of previously calibrated models, even when extrapolating the models to neighboring areas.

**Keywords:** proximal soil sensing; on-the-go data acquisition; soil fertility diagnostic; precision agriculture; digital soil mapping

## 1. Introduction

Site-specific management of soil fertility using precision agriculture (PA) approaches allows for the optimization of farm inputs, which increases profitability and sustainability of agricultural systems [1]. Soil fertility management based on mapping of its attributes is a common PA practice used all over the world [2,3]. This approach is especially important in Brazilian tropical soils that are naturally of low fertility and require the application of large amounts of fertilizer annually [4], which makes the country the fourth largest consumer of this input in the world [5]. About 15.3% of Brazilian soybean and corn farmers use this approach, totaling approximately 9 million hectares of mapped soils using the traditional technique that is based on a low spatial resolution sampling grid (e.g.,

$0.2-0.5$ sample ha$^{-1}$), traditional laboratory analyses, and geostatistics for data interpolation [6]. The low spatial resolution used in this traditional approach leads to unreliable mapping; it is a consensus that sampling grids larger than $100 \times 100$ m (<1 sample ha$^{-1}$) are inefficient in characterizing most soil fertility attributes [7,8].

For an accurate mapping of soil fertility attributes, a high spatial density of soil data acquisition is required (e.g., >1 sample ha$^{-1}$) [4,7,8]. However, this is not feasible using the traditional mapping technique due to costs related to laboratory analyses and time required for sample collection [9,10]. Alternatively, mobile platforms instrumented with proximal soil sensors acquire digital data about soil properties at fine scale through on-line measurements, i.e., with data acquisition performed on-the-go and at high frequency (e.g., one reading per second) [11,12]. This approach massively increases the spatial density of data acquisition (e.g., >50 data points ha$^{-1}$) [4]. Thereby, it allows one to describe the spatial continuity of soil attributes through geostatistical approaches improving the accuracy of mapping [13], especially for fertility attributes that present abrupt spatial changes (e.g., <100 m) [4,7]. In addition, this approach opens the possibility of performing automatic data collection, as well as diagnoses and interventions in real time [14].

In this context, visible and near-infrared spectroscopy (vis–NIR) is a technique that has attracted the attention of soil and biosystems engineering scientists for instrumentation of mobile platforms since it is compatible with on-line measurements [11]. Vis–NIR spectra provide information related primarily to soil mineral constituents (e.g., clay minerals), organic compounds (e.g., organic matter (OM) content), and water content [15], which are designated as primary soil properties [16]. In the spectrum, this information is represented by its intensity, shape, and absorptions (spectral features) at specific wavelengths [17]. The vis–NIR technique also allows estimation of soil properties with indirect relationships with the spectrum, such as pH, cation exchange capacity (CEC), base saturation (V), and exchangeable (ex-) nutrients; these attributes are designated as secondary soil properties [16]. This is possible when there is covariation of primary with secondary soil attributes [18]. Although there is vast literature showing the potential of on-line measurements using vis–NIR sensor for soil mapping in regions of temperate climate [14,16,19–23], tropical soils present different features, and such knowledge may not always be extrapolated [24]. The high temperature and moisture of the tropics promotes marked changes in mineralogical and biological characteristics of tropical soils when compared with soils of temperate regions, e.g., temperate soils usually feature more complex mineralogy than tropical soils and, in turn, biological activity increases notably in tropical soils compared to temperate ones [25]. These differences make it necessary to conduct studies to adapt and develop protocols for the use of mobile vis–NIR spectroscopy in the tropics. In addition, to the best of our knowledge, only one paper has explored this technique in Brazilian tropical soils [26].

The selection of an optimal data modeling method for predicting soil fertility attributes is a fundamental step toward accurate mapping of agricultural fields using mobile vis–NIR spectroscopy. This selection should take into account the predictive performance of the model, avoid procedures with excessive complexity, and ensure extrapolation at different spatial scales and environments [27,28]. Partial least-squares regression (PLS) model is a widely used method for its simplicity and robustness to deal with strongly collinear and noisy X-variables, a common characteristic of vis–NIR spectra [19,29,30]. PLS is a useful approach to model linear relationships between spectral variables and soil attributes [28]. However, modeling more complex and non-linear relationships may perform poorly using PLS [31]. Models involving artificial intelligence (AI) techniques, such as artificial neural network (ANN) models, may present superior performances to PLS for modeling soil attributes that are not directly related to vis–NIR spectra [28,29].

Numerous studies have shown the performance of different predictive models for digital soil mapping using mobile vis–NIR spectroscopy in soils of temperate regions [14,16,28,31–33]. Conversely, an accurate and reliable method for the estimation of soil fertility in Brazilian tropical soils has not been implemented to date and has been little explored in the litera-

ture [26]. This research aims to fill this gap, advancing the use of mobile vis–NIR spectroscopy for assessing key fertility attributes in tropical soils. The objectives of this study were (i) to investigate the feasibility of using mobile vis–NIR spectroscopy to predict and map key soil fertility attributes in a Brazilian agricultural field, and (ii) to compare prediction accuracy of PLS and ANN models for this purpose.

## 2. Materials and Methods

### 2.1. Study Area and Soil Sampling

The study area consists of a 138 ha field in the municipality of Campo Novo do Parecis, Mato Grosso State, Brazil (central coordinates: 14.102° S and 57.764° W, using WGS-84 datum) (Figure 1A,B), a region of tropical climate with wet summers and dry winters [34]. Before conducting the present study, this agricultural field was under a yearly crop rotation system with cotton and some grain crop (corn or soybean). The soil type is a Ferralsol [35], with texture varying between sandy loam and sand clay loam. The topography is rather flat with an elevation that oscillates between 607 and 630 m.

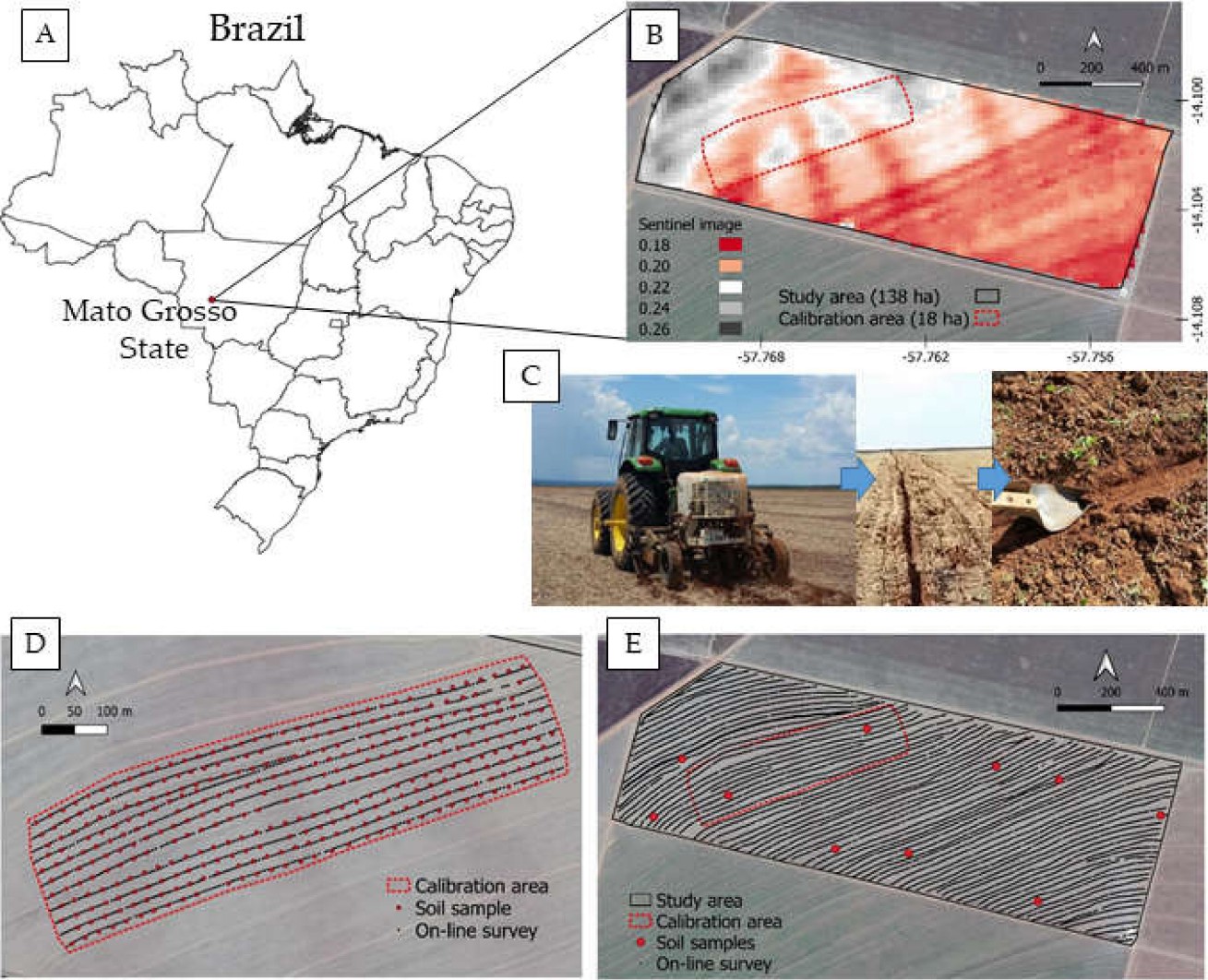

**Figure 1.** Schematic figure showing the location of the studied field (**A**); soil spatial variability observed in the Sentinel image (**B**); on-line soil survey conducted with the mobile vis–NIR platform and furrow left by the platform shank (**C**); soil sampling location and trajectory of on-line survey performed in the calibration area (**D**) and the entire field (**E**).

By using a SWIR-2 image from the satellite Sentinel 2-A (Band 12 which is centered at 2190 nm and has 20 m spatial resolution), an 18 ha area with high soil spatial variability was selected (Figure 1B) and designated as the calibration area. Prior to collecting the soil samples, the calibration area and the entire 138 ha field were scanned with the vis–NIR sensor in two separate on-line data acquisitions, respectively designated as spectral data acquisition 1 (Figure 1D) and 2 (Figure 1E). Spectral data were georeferenced using a global navigation satellite system (GNSS) receiver (StarFire, NavCom/John Deere, Torrance, CA, USA) with SF1 correction signal (NavCom/John Deere, Torrance, CA, USA), which provides an accuracy of about 0.5 m on the coordinates. After spectral data acquisition 1, 347 soil samples were collected in the calibration area, and after spectral data acquisition 2, 10 samples were collected in the entire field (designated in this study as verification points). Soil samples were collected in the furrows left by the platform shank (Figure 1C), using as reference the spectral data coordinates. In the calibration area, the average distance between each collected soil sample was 22 m in the row and 20 m between rows. The samples collected in the 138 ha field were positioned to cover most of the soil spatial variability observed in the spectral data. Both spectral data acquisitions were performed at the end of March (when the dry season starts) on consecutive days, without the occurrence of precipitation between them, such that soil moisture was low and with similar contents (about 5% $g^{-1}$).

In summary, first, the spectral data acquisition 1 was performed on the 18 ha area (designated as calibration area) followed by the collection of 347 soil in this same area (Figure 1D). This first dataset (spectral data acquisition 1 + soil analysis of the 347 soil samples) was used for the calibration and testing of the predictive models (as detailed in Section 2.4). Afterward, the spectral data acquisition 2 was performed on the entire 138 ha field (including a resampling of the calibration area), followed by the collection of the 10 soil samples (Figure 1E). This second dataset was used for independent testing of the previously calibrated models (as detailed in Section 2.5), the data from the first dataset were kept out of this new testing.

## 2.2. Mobile Platform and On-Line Vis–NIR Data Acquisition

On-line data acquisition was performed using the commercial mobile platform Veris® VisNIR (Veris Technologies, Salina, EUA) (Figure 2). It consists of a metal box with the vis–NIR system aligned with a sapphire window in its lower portion, attached to a subsoiler shank by a parallelogram mechanism to provide horizontality to the window (Figure 2B). The shank penetrates the soil, making a trench, where the optical unit acquire soil spectra from the smooth bottom of this trench. Fiber optic cables transmit the energy reflected from the soil to two spectrometers: an Ocean optics USB4000 (Ocean Optics Inc., Largo, FL, USA) and a Hamamatsu TG-cooled NIR-II (Hamamatsu Photonics K. K., Hamamatsu, Japan). The first one makes measurements from 373 to 1011 nm and the second one from 1170 and 2222 nm, both with spectral resolution around 5 nm. Spectral data are georeferenced using the GNSS system. The entire system was mounted onto the three-point linkage of a tractor.

Before data acquisition, the intensity of the light source was checked using four references materials with known spectral behavior. The system also self-calibrates periodically at each 10 min interval using a shutter system present inside the shank, which operates automatically. This periodic calibration is performed collecting a dark reference and a known internal reference material. The frequency of data acquisition was 1 Hz and each recorded spectrum corresponds to the average of readings collected during this interval. On-line measurements were carried out in transects relatively parallel, separated with about 20 m, at an average speed of 5 m $s^{-1}$, resulting in a spatial density of around 100 spectra $ha^{-1}$.

The acquired spectra presented noise in the intervals between 343–426, 989–1070 and 2153–2222 nm, which were removed for the subsequent analysis. The spectral behavior was descriptively analyzed to characterize its intensity, shape, and absorption features [36].

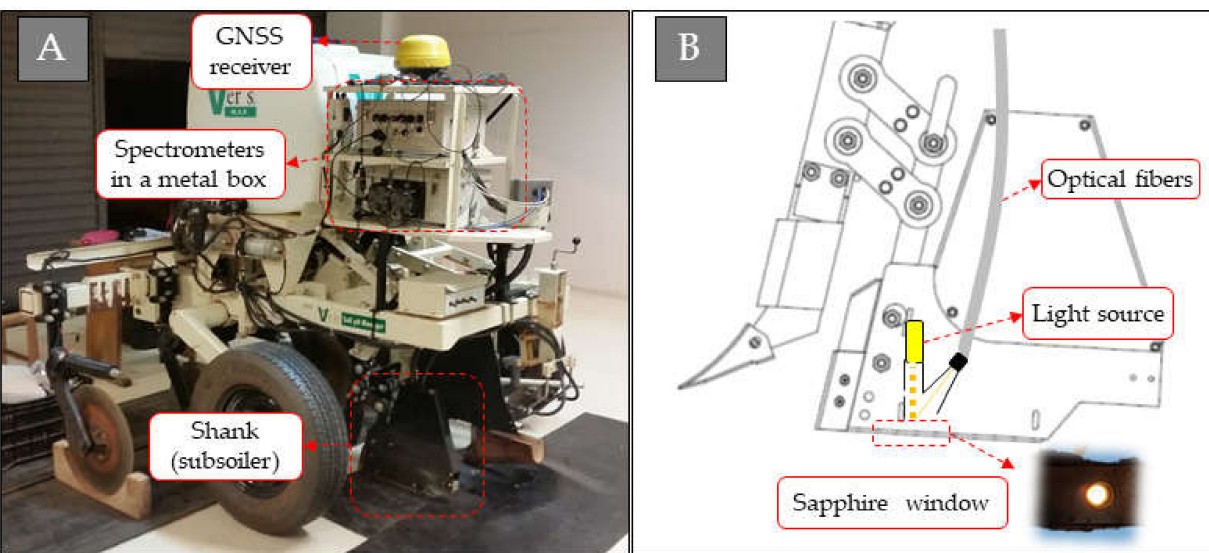

**Figure 2.** Mobile vis–NIR platform (**A**) and its shank in detail (**B**).

### 2.3. Laboratory Reference Analyses

Soil testing conducted in a commercial laboratory determined the contents of clay, OM, CEC, pH, V, ex-Ca, ex-Mg, ex-K, and ex-P, following the methodologies described by Van Raij [37]. Clay was quantified by the Bouyoucos hydrometer method. OM was determined via oxidation with potassium dichromate solution, and pH was determined via calcium chloride solution. Exchangeable nutrients were determined via ion exchange resin extraction. CEC was calculated by sum of the soil potential acidity and the bases ex-Ca, ex-Mg, and ex-K; for this, soil potential acidity was quantified via the buffer solution method (SMP). The results of these analyses were used as Y-variables for calibration and test of the models described in the following section.

### 2.4. Predictive Modeling Using Spectral Data Acquisition 1

The relationship between spectra (obtained from spectral data acquisition 1) and soil attributes measured in the 347 soil samples collected from the calibration area were established using partial least squares regression (PLS) and artificial neural network (ANN). The spectral data acquisition 2 and the 10 soil samples collected in the entire field were kept out of this analysis. The raw spectra, after removing intervals with noise, were used as X-variables.

The calibration and test of models were performed after subdividing dataset into two subsets of 85% (calibration/training set, n = 295) and 15% (test set, n = 52) using the Kennard–Stone algorithm performed on the measured fertility attributes. The Kennard–Stone algorithm is based on a random sample selection in which spectra from the original dataset are randomly assigned for training and testing. The algorithm uses Euclidean distance calculations to select the sample with maximum distance to all other samples, such that samples that are as far away from the selected samples as possible are selected, until the selected number of samples is reached. In this way, it is possible to avoid the presence of bias introduced manually or by a completely random data splitting [38]. For PLS models, the number of latent variables chosen was defined based on the maximum coefficient of determination ($R^2$) and lowest root-mean-square error (RMSE) obtained with leave-one-out cross-validation. The PLS method combines the useful spectral data into the first several latent variables which are then used for predictive modeling [39]; in other words, latent variables originate from a linear combination of the original variables. PLS models and dataset division were implemented using the Unscrambler software, version 10.5.1 (Camo AS, Oslo, Norway).

The ANN-based models were established with the Neural Network Toolbox in MAT-LAB R2016b (Mathworks Inc., Portola Valley, CA, USA) according to the fitting methodology. The ANN-based models have a Perceptron feedforward and multi-layered architecture, with a sigmoid transfer function in the hidden layer and a linear transfer function in the output layer. Different ANN architectures were run to fine-tune the hyperparameters (the number of layers, number of neurons, learning rate and momentum) with the supervised learning approach [33]. After fine tuning, performed by changing the hyperparameters in the sequence in which they were presented, the following settings were obtained: number of layers = 1 (simulation range: 1 to 5, step 1), number of neurons = 30 (simulation range 5 to 50, step 5), learning rate = 0.5 (simulation range: 0.1 to 0.7, step 0.1) and moment = 0.1 (simulation range: 0.1 at 0.7, step 0.1).

In order to evaluate the models prediction efficiency, $R^2$, RMSE, relative error (RMSE %), and the ratio of performance to interquartile range (RPIQ) [40] were evaluated. The following groups were used for RPIQ interpretation, as proposed by Nawar and Mouazen [32]: excellent models (RPIQ $\geq$ 2.5), very good models (2.5 > RPIQ $\geq$ 2.0), good models (2.0 > RPIQ $\geq$ 1.7), reasonable models (1.7 > RPIQ $\geq$ 1.4), and very poor models (RPIQ < 1.4). The best performing models were extrapolated to the data obtained from the on-line survey conducted on the entire area (spectral data acquisition 2) in order to map the 138 ha field. Their performances were again verified in terms of RMSE and RPIQ, using the 10 soil samples collected in the total area (as detailed below in the Section 2.5).

### 2.5. Model Test Using the Spectral Data Acquisition 2

In addition to the abovementioned test using 15% of the samples collected in the calibration area, an independent test using the spectral data acquired in the on-line survey conducted in the entire field was performed. The purpose of this independent test was to verify the reliability and the extrapolability of the calibrated models when applied in spectral data that were not used for model calibration. Only the best performing models were applied in the spectral data, and the 10 soil samples (verification points) were used to attest their predictive accuracy through their performance of RMSE and RPIQ.

## 3. Results

### 3.1. Laboratory Measured Soil Properties

Figure 3 shows the box plot and coefficient of variation (CV) of soil attributes for the calibration and test datasets. All attributes have a comparable range and CV for the sets used to calibrate and validate the models, which were guaranteed by splitting the data using the Kennard–Stone algorithm. It is a desired feature to ensure that the observed prediction quality is related to sensor performance and not over- or underestimated due to diverging characteristics in the dataset [12,41].

The correlation matrix of the different soil attributes is important to interpret possible indirect determinations, i.e., predictions of attributes that are not directly related to vis–NIR spectra (e.g., CEC, V, pH, and exchangeable nutrients) [18]. In this sense, significant correlations between CEC, V, ex-Ca, ex-Mg, and ex-P with OM, as well as, between CEC, ex-Mg, ex-K, and ex-P with clay can be highlighted (Table 1).

The spatial statistics presented in Table 1 show that all fertility attributes presented spatial dependence ranging from moderate to strong (38.90 $\geq$ SDD $\geq$ 8.40), and therefore amenable to being modeled and mapped [13]. The ex-P is the only exception, since it showed a pure nugget effect. It was also observed that the range of spatial dependence of clay, OM, and CEC reached higher values ($\geq$100 m), while pH, V, ex-Ca, ex-Mg, and ex-K showed more abrupt variations (range $\leq$ 56 m).

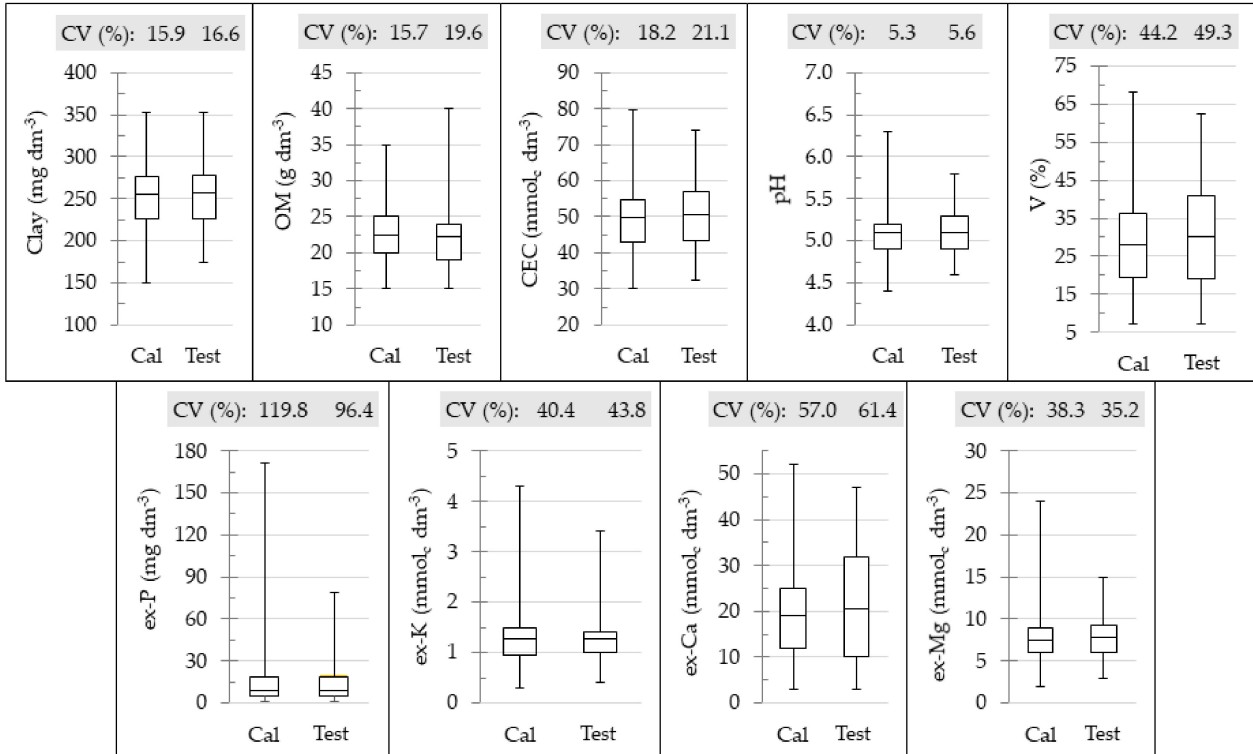

**Figure 3.** Box plots presenting the variation for the contents of clay, organic matter (OM), cation exchange capacity (CEC), pH, base saturation (V), and extractable (ex-) P, ex-K, ex-Ca, and ex-Mg for each calibration (Cal) and test datasets. The coefficient of variation (CV) is also presented above each box plot.

**Table 1.** Spatial statistics and correlation matrix for the 347 samples collected in the calibration area. Soil sampling was carried out at a spatial density of about 20 samples ha$^{-1}$.

| | Clay | OM [1] | CEC [2] | pH | V [3] | ex-Ca [4] | ex-Mg [4] | ex-K [4] | ex-P [4] |
|---|---|---|---|---|---|---|---|---|---|
| **Spatial Statistics:** | | | | | | | | | |
| Nugget effect | 173.10 | 6.65 | 33.00 | 0.01 | 30.00 | 20.00 | 2.80 | 0.10 | PNE [7] |
| Sill | 2050.20 | 17.09 | 89.50 | 0.07 | 296.90 | 127.30 | 8.10 | 0.30 | PNE [7] |
| Range [5] | 176 | 113 | 100 | 35 | 55 | 56 | 47 | 42 | PNE [7] |
| SDD [6] * | 8.40 | 38.90 | 36.90 | 14.30 | 10.10 | 15.70 | 34.80 | 32.30 | - |
| **Correlation Matrix:** | | | | | | | | | |
| Clay | 1.00 | | | | | | | | |
| OM | **0.27** | 1.00 | | | | | | | |
| CEC | **0.35** | **0.30** | 1.00 | | | | | | |
| pH | −0.01 | 0.10 | **0.18** | 1.00 | | | | | |
| V | 0.08 | **−0.17** | **0.56** | **0.50** | 1.00 | | | | |
| ex-Ca | 0.01 | **0.13** | **0.17** | **−0.14** | −0.08 | 1.00 | | | |
| ex-Mg | **0.20** | **0.14** | **0.19** | −0.08 | −0.06 | **0.37** | 1.00 | | |
| ex-K | **0.16** | −0.08 | **0.80** | **0.31** | **0.90** | 0.01 | −0.06 | 1.00 | |
| ex-P | **0.17** | **0.22** | **0.43** | **0.69** | **0.52** | 0.04 | **−0.11** | 0.08 | 1.00 |

[1] Organic matter; [2] cation exchange capacity; [3] base saturation; [4] exchangeable (ex-) nutrients; [5] range values expressed in meters; [6] spatial dependence degree; [7] pure nugget effect. Bold values indicate a significant correlation (*p* value < 0.05); * SDD values can be interpreted as strong spatial dependence when less than 25%, moderate spatial dependence when between 25 and 75%, and weak spatial dependence when greater than 75% [13].

### 3.2. Descriptive Analysis of Vis–NIR Spectra

Changes in soil physical, chemical, and mineralogical properties can influence the behavior of vis–NIR spectra, with reflections on (i) their overall intensity (albedo), (ii) the intensity and amplitude of absorption features, and (iii) the shape of the spectrum [36]. The main spectral features observed in this study (Figure 4), are: alterations of albedo; presence of iron oxide/hydroxide (Fe]-OH) features at 425, 650 and 903 nm; and hydroxyl group (O-H) features at 1400 and 1900 nm, which are related to the structure of water + 1:1 and water + 2:1 minerals, respectively [18].

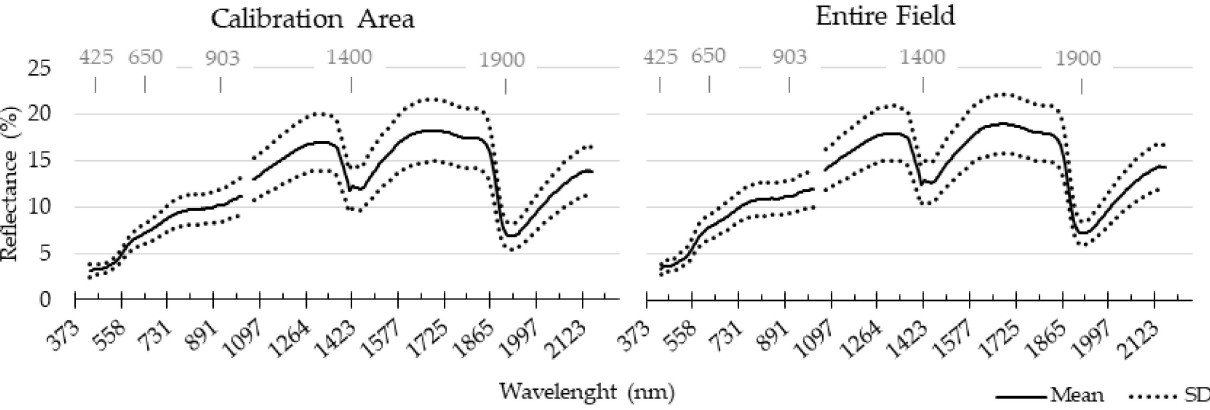

**Figure 4.** Mean and standard deviation (SD) of visible and near-infrared (vis–NIR) spectra acquired through the on-line survey performed in both the calibration area (spectral data acquisition 1) and the entire field (spectral data acquisition 2).

In Figure 4, it is also possible to observe the similarity of the mean and standard deviation of the spectra obtained in both the calibration area and the entire field. This behavior emphasizes the representativeness of the calibration area to cover the spectral variation of the study area as a whole.

### 3.3. Predictive Performance of Mobile Vis–NIR Spectroscopy in the Calibration Area

The predictive models for clay, OM, CEC, V, and ex-Ca showed satisfactory performance in their test using both ANN and PLS models, with RPIQ oscillating between 1.4 and 2.4 for PLS, and varying between 1.7 and 2.7 with ANN (Table 2). The ANN prediction accuracy in the test set was higher than the PLS for all attributes predicted with satisfactory performance (clay, OM, CEC, V, and ex-Ca) (Table 2). The RMSE values reduced 8.1%, 36.4%, 2.4%, 19.1% and 1.2% for the prediction of clay, OM, CEC, V, and ex-Ca, respectively, when using ANN instead of PLS models (Table 2). In addition, when using ANN models, predictive performances with RPIQ $\geq$ 2.0 (qualified as very good and excellent) were obtained for clay, OM, V, and ex-Ca. In turn, when using PLS, predictive performances with RPIQ $\geq$ 2.0 were obtained just for clay and V. The prediction of pH, ex-Mg, ex-K, and ex-P showed poor performance (RPIQ $\leq$ 1.2) in the test of all evaluated models.

**Table 2.** Prediction results of the calibration and test set obtained from partial least squares regression (PLS) and artificial neural network (ANN) models calibrated using mobile visible and near-infrared spectroscopy (vis–NIR) in the calibration area.

|  | Clay | OM [1] | CEC [2] | pH | V [3] | ex-Ca [4] | ex-Mg [4] | ex-K [4] | ex-P [4] |
|---|---|---|---|---|---|---|---|---|---|
| **ANN Calibration (n = 295):** | | | | | | | | | |
| $R^2$ | 0.89 | 0.66 | 0.69 | 0.32 | 0.82 | 0.76 | 0.48 | 0.50 | 0.29 |
| RMSE | 13.15 | 1.78 | 4.15 | 0.23 | 7.11 | 5.51 | 2.07 | 0.39 | 14.46 |
| RMSE % | 6.48 | 11.12 | 8.36 | 12.22 | 9.61 | 11.24 | 9.41 | 11.40 | 8.71 |
| RPIQ | 3.9 | 2.8 | 2.9 | 1.3 | 3.9 | 2.9 | 1.4 | 1.3 | 0.6 |
| **ANN Test (n = 52):** | | | | | | | | | |
| $R^2$ | 0.77 | 0.57 | 0.55 | 0.10 | 0.65 | 0.69 | 0.23 | 0.14 | 0.12 |
| RMSE | 19.89 | 2.32 | 7.24 | 0.24 | 10.27 | 7.14 | 2.92 | 0.55 | 18.11 |
| RMSE % | 9.80 | 14.53 | 14.60 | 12.84 | 13.88 | 14.57 | 13.25 | 16.21 | 10.91 |
| RPIQ | 2.6 | 2.2 | 1.7 | 1.2 | 2.7 | 2.2 | 1.0 | 0.9 | 0.4 |
| **PLS Calibration (n = 295):** | | | | | | | | | |
| $R^2$ | 0.76 | 0.48 | 0.41 | 0.01 | 0.59 | 0.69 | 0.06 | 0.04 | 0.01 |
| RMSE | 20.06 | 2.53 | 6.98 | 0.27 | 10.37 | 6.11 | 2.79 | 0.51 | 20.47 |
| RMSE % | 9.88 | 12.67 | 14.07 | 14.26 | 14.01 | 12.47 | 12.69 | 12.73 | 12.33 |
| RPIQ | 2.5 | 2.0 | 1.7 | 1.1 | 2.6 | 2.1 | 1.1 | 1.1 | 0.4 |
| n VL | 9 | 8 | 9 | 1 | 11 | 14 | 3 | 1 | 1 |
| **PLS Test (n = 52):** | | | | | | | | | |
| $R^2$ | 0.75 | 0.29 | 0.52 | 0.00 | 0.49 | 0.67 | 0.08 | 0.01 | 0.03 |
| RMSE | 21.64 | 3.65 | 7.42 | 0.28 | 12.71 | 7.23 | 2.60 | 0.55 | 15.84 |
| RMSE % | 10.66 | 18.24 | 14.96 | 14.84 | 17.18 | 14.76 | 11.83 | 13.87 | 9.54 |
| RPIQ | 2.4 | 1.4 | 1.6 | 1.1 | 2.1 | 1.8 | 1.2 | 1.0 | 0.6 |
| n VL [5] | 9 | 8 | 9 | 1 | 11 | 14 | 3 | 1 | 1 |

[1] Organic matter; [2] cation exchange capacity; [3] base saturation; [4] exchangeable (ex-) nutrients; [5] number of latent variables The coefficient of determination ($R^2$) and ratio of performance to interquartile range (RPIQ) values are presented on grayscale, highlighting the highest values. The root-mean-square error (RMSE) was given in g dm$^{-3}$ for clay and OM; in mmol$_c$ dm$^{-3}$ for CEC, ex-K, ex-Ca, and ex-Mg; in % for V; and in mg dm$^{-3}$ for ex-P.

### 3.4. Prediction Performance of the Independent Test (Using the Spectral Data Acquisition 2)

The predictive models for clay, OM, CEC, V, and ex-Ca that performed satisfactorily in the calibration area (Table 2) were transferred to the spectra of the second on-line data acquisition, and their performances are shown in Figure 5. The results showed that only the clay and the OM prediction models were reliable (with RPIQ = 2.49 for both attributes) when extrapolated to the entire field. Predictions of CEC, V, and ex-Ca showed very poor performance, with RPIQ $\leq$ 1.08, and a scatter plot of measured versus predicted values with points completely outside the 1:1 reference line (Figure 5E,G,I).

The range and dispersion of clay, OM, CEC, V, and ex-Ca attributes of the calibration and independent test datasets are shown in Figure A1. Although clay has a higher upper limit of the independent test dataset than that of the calibration set (which is not recommended in predictive analyses), the relation between spectra and clay did not change in this extrapolated concentration range (i.e., between 352 and 482 mg dm$^{-3}$) such that the calibrated model maintained its good performance. Conversely, although the CEC and ex-Ca attributes presented ranges compatible with those observed in the calibration set, their predictive models did not perform well, indicating that the spectra changed the relationship with these attributes when considering the entire field.

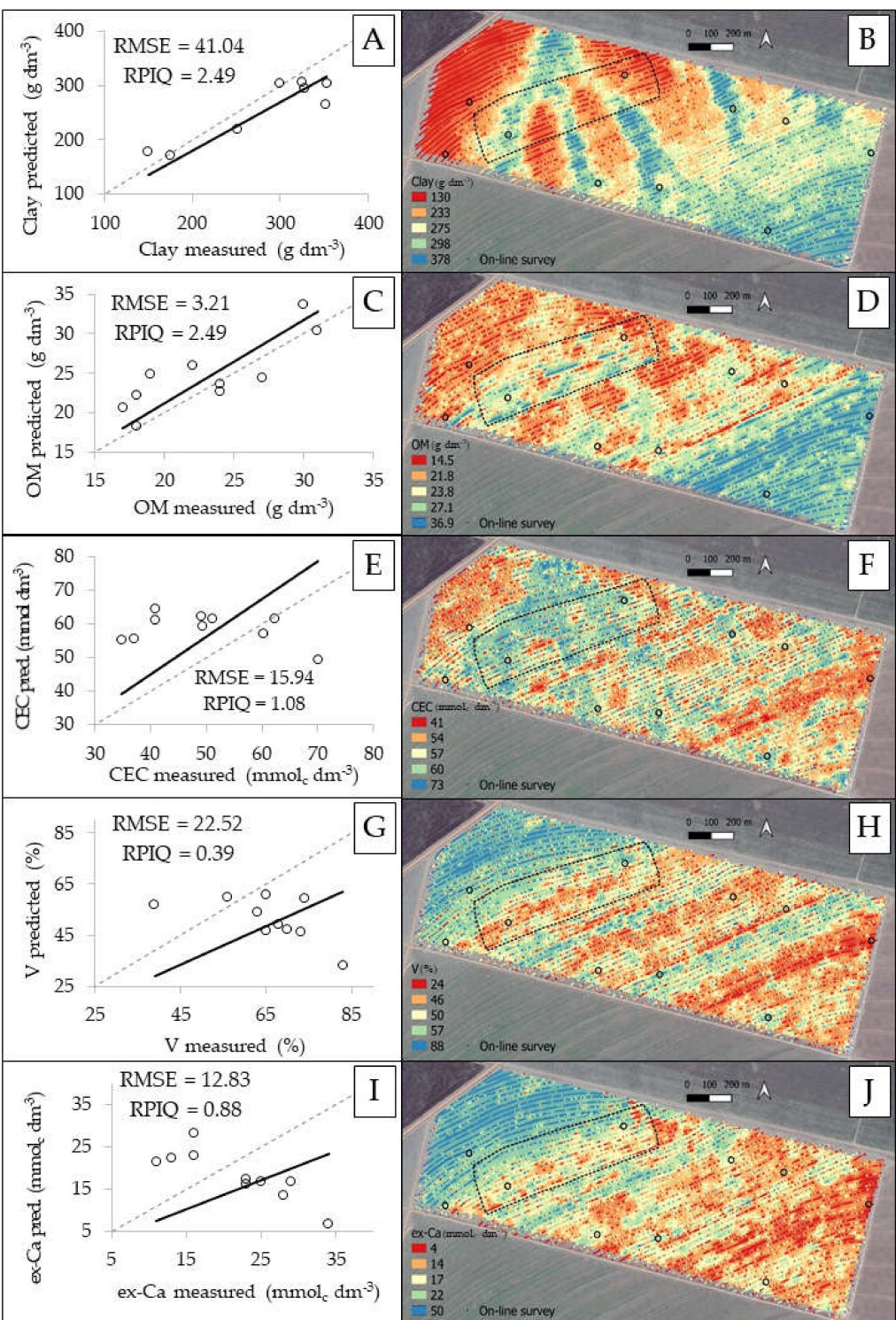

**Figure 5.** Scatter plots (for the 10 verification points) of measured versus predicted fertility attributes that presented satisfactory performance (in the calibration area) while using artificial neural network (ANN) models (**A,C,E,G,I**); the prediction results of this independent testing are detailed in Table A1 (Appendix A). Fertility attribute maps obtained after interpolation of the predictions using ANN models (**B,D,F,H,J**). The data were interpolated by kriging using the Vesper 1.63 software [42] and a 10 × 10 m grid, i.e., with a resolution compatible with that of the data acquisition. On the maps, the boundary of the calibration area is represented in black dashed lines and the position of the soil samples used in the independent testing in black circles.

## 4. Discussion

In geostatistics, the range of an attribute configures the distance in which samples present spatial autocorrelation. It is a consensus that for a reliable mapping of soil attributes it is necessary that the sampling grid should have a distance between samples of at least half the range of the target attribute [13]. In other words, sampling in grids with a distance superior to the range of the target attribute is not sufficient for its spatial characterization, producing unreliable maps. Thus, considering the range of fertility attributes presented in Table 1 ($35 \leq$ range $\leq 176$), the spatial resolution performed by the mobile vis–NIR platform ($20 \times 5$ m, resulting in 100 data ha$^{-1}$) was sufficient for mapping of all the attributes of interest. The only exception was the ex-P, which was not possible to obtain its range, since in the geostatistical analysis using 20 samples ha$^{-1}$ it showed pure nugget effect. Although ex-P mappings have been successfully performed on soils in temperate regions with on-line surveys using vis–NIR platforms ($R^2 = 0.60$) [19], tropical soils show strong adsorption of this nutrient by Fe oxides [25], which confers abrupt spatial variations to this nutrient due to its low mobility [4].

In the calibration area, predictive models with satisfactory performance (RPIQ $\geq 1.7$) were obtained for clay ($R^2 = 0.77$), OM ($R^2 = 0.57$), CEC ($R^2 = 0.55$), V ($R^2 = 0.65$), and ex-Ca ($R^2 = 0.69$). Accurate predictions ($0.73 \leq R^2 \leq 0.90$) of clay and OM using mobile vis–NIR spectroscopy are commonly reported in the literature for soils from temperate regions [21,28,33]. Predictions of textural contents via vis–NIR spectra are justified by two factors: (i) the relationship between the albedo and the sample grain size, and (ii) the presence of clay mineral features, such as those observed at 1400 and 1900 nm [15,43]. In turn, OM is related to smoothing of features and modifications to the shape of the spectrum, making it more straight and concave mainly in the visible region [36,44]. Predictions of secondary attributes through vis–NIR data are less common [18], but have also been reported in several other studies in temperate soils, such as by Munnaf et al. [16] for pH, ex-Na, ex-Ca, ex-Mg, ex-K ($0.58 \leq R^2 \leq 0.81$), by Mouazen and Kuang [19] for ex-P ($R^2 = 0.60$), and by Ulusoy et al. [45] for CEC ($0.70 \leq R^2 \leq 0.75$). Although these are good results, in tropical Brazilian soils, the only study found that used on-line vis–NIR spectroscopy obtained reasonable and poor performances for the prediction of key soil fertility attributes (e.g., CEC, V, pH, soil potential acidity, ex-Ca, and ex-Mg), with $R^2$ oscillating between $-0.01$ and $0.40$, with the pH model as the one that presented the best performance ($R^2$ of 0.40 and RPIQ of 2.30) [26]. In general, predictions of secondary attributes, mainly macronutrients, are still a challenge worldwide [4]. The automation of soil mapping using mobile platforms or robots with multi-sensor systems (e.g., combining vis–NIR and X-ray fluorescence sensors) is an alternative to increase the number of soil fertility attributes predicted satisfactorily, since the fusion of data from these sensors has shown promising results for the prediction of key soil fertility attributes [46,47].

Our results showed a better prediction performance for models calibrated with ANN compared to PLS. For example, in the calibration area, predictive models for clay and OM showed, respectively, RMSE values (of the test set) 8.7 and 57.3% higher when using PLS instead of ANN models. Better predictive performances for ANN models compared to PLS have been reported by Kuang et al. [33], which showed RMSE reduction ranging from 2.6 to 5.2% for clay, organic carbon (OC), and pH prediction. Slightly higher predictive performances for ANN models compared to PLS have also been reported by Fariftech et al. [29] for modeling soil spectral data. Alterations in the vis–NIR spectrum caused by changes in soil properties are often subtle and do not always present a linear pattern, such that the predictive performance of predictive models changes depending on the ability of the algorithm to extract useful information [48,49]. Vis–NIR spectra obtained directly in the field suffer interference from external factors, such as variations in soil moisture and structure, these factors contribute to making the relationship between spectra and soil attributes more complex, which may justify the superior performance of non-linear ANN models. Non-linear models are being proposed in the literature for the development of predictive models using spectral data, assuming that they are solid alternatives to exploring

hidden and non-linear spectral information [22,50]. In this study, we observed that although it is possible to obtain satisfactory predictions for some attributes using PLS, the optimal approach was modeling via ANN.

The inference about secondary attributes has an indirect nature [10]; hence, some care should be taken to not overstate the extrapolability of predictive models [51]. In our study, prediction models for CEC, V, and ex-Ca showed satisfactory performance in the calibration area but performed poorly when used in the entire field, indicating that the spectra changed the relationship with these attributes when considering the entire field. Conversely, the clay and OM models repeated the good performances observed in the calibration area when evaluated in the entire field. This behavior emphasizes the potential of using mobile vis–NIR spectroscopy for clay and OM mapping at a fine scale in Brazilian tropical soils. Conversely, secondary attribute predictions, although possible, need care for the extrapolation of their models. It also highlights the need to implement verification points to assess reliability and extrapolability of previously calibrated models, even for application in areas neighboring the site where the models were calibrated. Kuang et al. [33] observed a significant reduction in the performance of predictive models for clay, OC, and pH (with $R^2$ values reducing from 0.71–0.86 to 0.37–0.55) when applying an independent testing (i.e., without using local spectra in the model calibration) using vis–NIR data collected on-line in two Danish agricultural fields. Despite this, the authors still obtained satisfactory performance (with residual prediction deviation (RPD) $\geq$ 1.4) for clay prediction in both fields and for OC prediction in one of the fields. The predictive performances observed in the present study for the independent testing of clay and OM models were superior to those reported by Kuang et al. [33], which may be explained by the fact that our calibration was performed in a neighboring area, whereas the other authors used data from a spectral library for calibration.

Simplifying the calibration process of predictive modeling using mobile vis–NIR sensors is key to automating and expanding the mapping of soil attributes [52]. Spectra acquired directly in the field may be affected by external factors, such as moisture and granulometric variations, making it necessary in some cases to apply methods to mitigate these factors in order to obtain accurate predictive models [53,54]. However, the use of these methods makes the calibration process more complex. In our study, excellent predictive models were obtained for clay and OM (in the independent test, Figure 5) only with local calibrations using ANN and without the application of methods for mitigating external factors. It is possible that this performance was obtained due to low soil moisture content during data collection (about 5% g$^{-1}$), which is a common feature in Brazilian tropical soils during the dry season from April to September. This hypothesis needs to be verified in future studies and, if proven, the use of methods to mitigate external factors may be neglected in on-line spectral acquisitions performed in relatively dry soils. Finally, it is also suggested that further studies explore new generalist calibration methods (e.g., using regional and national spectral libraries and spiking techniques [32]) that allow calibration of predictive models compatible with larger scale applications (e.g., agricultural areas with spectrally compatible soils).

## 5. Conclusions

Mobile visible and near-infrared spectroscopy (vis–NIR) allowed for the acquisition of data with adequate spatial resolution (i.e., compatible with the range of spatial variation) for mapping eight out of the nine key soil fertility attributes assessed, these being clay, organic matter (OM), cation exchange capacity (CEC), pH, base saturation (V), and the exchangeable (ex-) nutrients ex-Ca, ex-Mg, and ex-K. However, predictive models with satisfactory performance (ratio of performance to interquartile range (RPIQ) $\geq$ 1.4) were obtained only for clay, OM, CEC, V, and ex-Ca, in the calibration area (18 ha area), and only for clay and OM (RPIQ = 2.49, for both) when performing an independent test, i.e., extrapolating these models to the entire field (138 ha field).

Artificial neural network (ANN) models showed superior performance versus partial least squares regression (PLS) models for all evaluated attributes. Local models calibrated with ANN combined with on-line survey conducted during the Brazilian dry season showed robust predictions (reliable and replicable) for clay and OM, even without using methods to mitigate external factors (e.g., soil moisture).

This study shows the potential of using mobile vis–NIR spectroscopy for mapping clay and OM in Brazilian agricultural fields, indicating the need to use an independent test to assess the performance of local models previously calibrated. Further research is needed to explore different datasets acquired from a larger number of fields with different soil types, textural characteristics, and agricultural practices. Nevertheless, it confirms that mobile vis–NIR systems operating directly in the field can provide successful mappings considering the element in analysis.

**Author Contributions:** Conceptualization, M.T.E. and J.P.M.; Methodology, M.T.E., T.R.T., R.V.d.S. and J.P.M.; Validation, T.R.T., R.G.T., J.A.M.D. and J.P.M.; Formal analysis, T.R.T., M.T.E. and R.G.T.; Investigation, T.R.T., M.T.E. and J.P.M.; Resources, M.T.E. and J.P.M.; Data curation, T.R.T., R.V.d.S. and R.G.T.; Writing—original draft preparation, T.R.T.; Writing—review and editing, M.T.E., R.V.d.S., R.G.T., J.A.M.D. and J.P.M.; Visualization, T.R.T., M.T.E. and J.P.M.; Supervision, J.A.M.D. and J.P.M.; Project administration, M.T.E. and J.P.M.; Funding acquisition, M.T.E. and J.P.M. All authors have read and agreed to the published version of the manuscript.

**Funding:** T.R.T. and M.T.E. were funded by the São Paulo Research Foundation (FAPESP), grant number 2020/16670-9 and 2014/10737-3, respectively. Soil fertility tests and field activities were funded by CNPq—"Edital de Chamada Universal", grant number 458180/2014-9. The Funding Authority for Studies and Projects (FINEP) for the support to PROSENSAP project that allowed the acquisition of the mobile vis–NIR platform.

**Institutional Review Board Statement:** Not applicable.

**Informed Consent Statement:** Not applicable.

**Acknowledgments:** We thank the company Terra Santa Agro S.A. for ceding the study area and providing full support in the field data survey.

**Conflicts of Interest:** The authors declare no conflict of interest.

## Appendix A

**Table A1.** Prediction results of the independent test set obtained from artificial neural network (ANN) models calibrated using mobile visible and near-infrared spectroscopy (vis–NIR).

|  | **Clay** | **OM** [1] | **CEC** [2] | **V** [3] | **ex-Ca** [4] |
|---|---|---|---|---|---|
| $R^2$ | 0.83 | 0.68 | 0.11 | 0.00 | 0.00 |
| RMSE | 41.04 | 3.21 | 15.94 | 22.52 | 12.83 |
| RMSE % | 14.70 | 13.97 | 32.14 | 34.33 | 58.87 |
| RPIQ [5] | 2.49 | 2.49 | 1.08 | 0.39 | 0.88 |

[1] Organic matter; [2] cation exchange capacity; [3] base saturation; [4] exchangeable Ca; [5] ratio of performance to interquartile range. The root-mean-square error (RMSE) was given in g dm$^{-3}$ for clay and OM; in mmol$_c$ dm$^{-3}$ for CEC and ex-Ca; and in % for V.

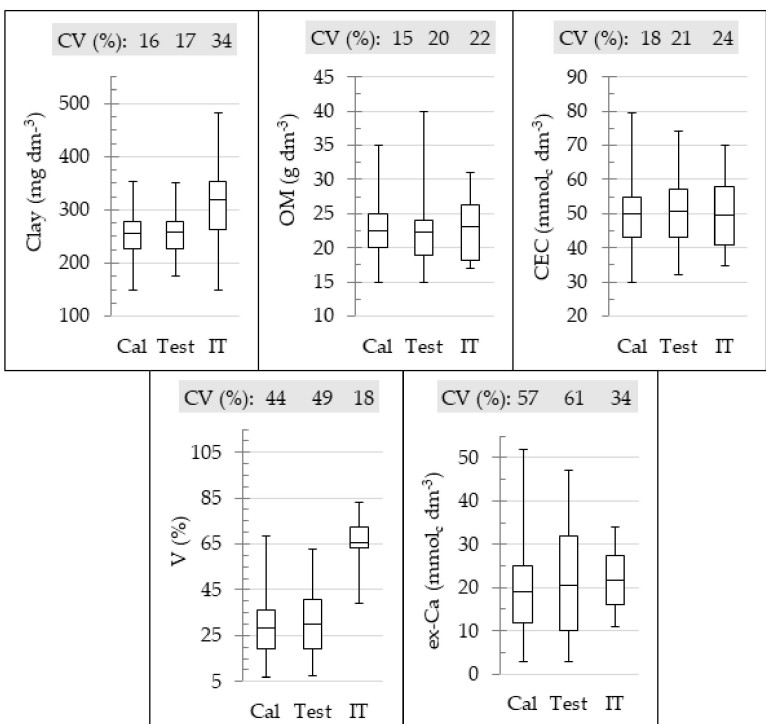

**Figure A1.** Box plots presenting the variation for the contents of clay, organic matter (OM), cation exchange capacity (CEC), base saturation (V), and extractable (ex-) Ca for the calibration (Cal), test, and independent test (IT) datasets. The coefficient of variation (CV) is also presented above each box plot.

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
