# Peer review of "Predictive Performance of Mobile Vis–NIR Spectroscopy for Mapping Key Fertility Attributes in Tropical Soils through Local Models Using PLS and ANN"

_2673-4052, doi:10.3390/automation3010006_

Round 1

Reviewer 1 Report

In this study, two modeling methods, partial least squares algorithm and artificial neural network algorithm, were used to develop a prediction model of soil fertility in a Brazilian agricultural field. This model has good predictive ability. However, there are still some areas for improvement in this paper.

(1) Introduction section does not sufficiently describe the necessity, novelty and expected contributions of the work. Please present a more specific, structural literature review associated with the background, the state-of-art of research topics/methods/societal needs, the relevance and objectives. I suggest that authors can add some new papers published in recent years to their references to reflect the highlights of your paper as well as its novelty.

(2) Authors need to ensure the clarity of the illustrations in the text and are advised to use high-resolution illustrations in the text.

(3) Section 2.4 Predictive modelling, authors did not specifically introduce how to divide the training set and test set, it is recommended to add the details of data set division.

(4) Section 2.4 Predictive modelling, authors mention running different ANN architectures to fine-tune the hyperparameters, but the authors do not specify how the fine-tuned hyperparameters are set or what the criteria for fine-tuning the hyperparameters are. I suggest that the model building section be described in detail.

(5) Section 4 Discussion, “Our results showed a better prediction performance for models calibrated with ANN compared to PLS”, authors briefly analyze the reasons for this phenomenon, but I think the analysis lacks depth. The author just mentions that the predictive performance of predictive models changes depending on the ability of the algorithm to extract useful information, but they do not explain why the ANN model has a stronger ability to extract useful information. I suggest that the authors discuss this in detail.

All in all, the analysis of the article is not deep enough, a great deal of research work has already been done in this research direction. The author should revise it thoroughly and resubmit it.

Author Response

Thank you for carefully reading our study and for provide us insightful comments to improve the quality of the paper. The authors tried their best to improve to the original manuscript and fulfil the reviewer suggestions. Based on your comments, the manuscript was revised using the track changes function activated in Word.

Below you can find our responses to each of your comments.

  1. Comments: “(1) Introduction section does not sufficiently describe the necessity, novelty and expected contributions of the work. Please present a more specific, structural literature review associated with the background, the state-of-art of research topics/methods/societal needs, the relevance and objectives. I suggest that authors can add some new papers published in recent years to their references to reflect the highlights of your paper as well as its novelty.”

Answer:  Thank you for this important comment that helped us improve our introduction. We have added new information to the introduction (please see L44-46, L50-52, L58-63, and L79-88) seeking to make clearer the necessity, novelty and contribution of the paper. We also added some more recent papers in the introduction, which were further discussed in the discussion. Finally, we would like to comment that the main motivation for the development of our study is the lack of literature addressing the application of mobile vis-NIR spectroscopy in Brazilian tropical soils. Only one study has been carried out in this context. Furthermore, tropical soils are markedly different from temperate soils - where the technology was developed -, which prevents us from simply extrapolating the related knowledge.

  1. Comments – “(2) Authors need to ensure the clarity of the illustrations in the text and are advised to use high-resolution illustrations in the text.”.

Answer: Ok. We have changed all the figures to increase their resolution.

  1. Comments – “(3) Section 2.4 Predictive modelling, authors did not specifically introduce how to divide the training set and test set, it is recommended to add the details of data set division.”.

Answer: We have added more information about the data set division (please see L201-209). The calibration/training and test of models were done after subdividing data set into two subsets of 85% (calibration/training set, n = 295) and 15% (test set, n = 52) using the Kennard–Stone algorithm performed on the measured fertility attributes. This algorithm was used to ensure that all attributes have comparable range and coefficient of variation for the sets used to calibrate and validate the models; in turn, this ensures that the observed prediction quality is related to sensor performance and not over- or underestimated due to diverging characteristics in the dataset. This information was presented in L250-253, in the Results Section.

  1. Comments – “ (4) Section 2.4 Predictive modelling, authors mention running different ANN architectures to fine-tune the hyperparameters, but the authors do not specify how the fine-tuned hyperparameters are set or what the criteria for fine-tuning the hyperparameters are. I suggest that the model building section be described in detail”.

Answer: Thank you for the suggestions. We have added new information to the manuscript seeking to improve the explanation with greater detail of the technique used (please, see L223).

  1. Comments – (5) Section 4 Discussion, “Our results showed a better prediction performance for models calibrated with ANN compared to PLS”, authors briefly analyze the reasons for this phenomenon, but I think the analysis lacks depth. The author just mentions that the predictive performance of predictive models changes depending on the ability of the algorithm to extract useful information, but they do not explain why the ANN model has a stronger ability to extract useful information. I suggest that the authors discuss this in detail”.

Answer: Thank you for this comment. Different studies conducted on temperate soils (see references below) have obtained these results, i.e. superior predictive performance for ANN compared to PLS. However, as commented by Kuang et al. (2015), there is still no consensus on the reason why ANN provides a better performance. In our study, we believe that a good reason for this is that external factors (e.g., alterations in soil moisture and structure, which are present in data acquired directly in the field) interfere with the relationship between spectra and fertility attributes, making it more complex; hence, non-linear models are required. We have added this explanation in L385, as you can also follow below:

“Vis-NIR spectra obtained directly in the field suffer interference from external factors, such as variations in soil moisture and structure, these factors contribute to make the relationship between spectra and soil attributes more complex, which may justify the superior performance of non-linear ANN models. Non-linear models are being pro-posed in the literature for the development of predictive models using spectral data assuming that they are solid alternatives to explore hidden and non-linear spectral in-formation”.

# Studies reporting superior predictive performance for ANN compared to PLS:

1 - Mouazen, A.M., Kuang, B., De Baerdemaeker, J., Ramon, H., 2010. Comparison between principal components: partial least squares and artificial neural network analyses for accuracy of measurement of selected soil properties with visible and near infrared spectroscopy. Geoderma 158, 23–31.

2 - Stenberg, B., 2010. Effects of soil sample pretreatments and standardised rewetting as interacted with sand classes on Vis-NIR predictions of clay and soil organic carbon. Geoderma 158 (1–2), 15–22.

3 - Tekin, Y., Tumsavas, Z., Mouazen, A.M., 2012. Effect of moisture content on prediction of organic carbon and pH using visible and near infrared spectroscopy. Soil Sci. Soc. Am. J. 76 (1), 188–198.

4 - Viscarra Rossel, R.A., Behrens, T., 2010. Using data mining to model and interpret soil diffuse reflectance spectra. Geoderma 158, 46–54.

4 - Kuang, B., Tekin, Y. and Mouazen, A.M., 2015. Comparison between artificial neural network and partial least squares for on-line visible and near infrared spectroscopy measurement of soil organic carbon, pH and clay content. Soil and Tillage Research, 146, pp.243-252.

Finally, we would like to comment that we truly appreciate all your comments and we really made efforts to handle your concerns. We do hope the revised manuscript deserves your recommendation for Automation. 

If necessary, we continue disposed to make other alterations.

Sincerely,

The authors.

Reviewer 2 Report

The authors of Predictive performance of mobile Vis-NIR spectroscopy for mapping key fertility attributes in tropical soils through local models using PLS and ANN combine field data detailing soil characteristics and information derived via visible and near-infrared spectroscopy with two types of models to predict and map key fertility attributes in tropical soils. I found this work interesting. Overall I believe this paper would be interesting to other researchers in this field and that the methodologies and results presented appear rigorous. I have provided several suggestions below which I believe could improve the manuscript. My one major concern is that the article contains a number of grammar issues. I have pointed out the first 5 instances of such issues below.

Line by line comments:

Line 58: Defining on-line measurements could be useful as this article could be of interest to readers from different scientific backgrounds.

Lines 83-86: The main novel aspect of this paper is that the methods are being applied in Brazilian tropical soils rather than temperate regions. Some additional context discussing why this region is important and why these methodological developments are necessary could improve the introduction.

Lines 168-170: I suggest editing this text and maybe also figure 1 for clarity to emphasize that the model was trained using the on-line surveys and 347 soil samples in the calibration area and that both the on-line survey data (which is a repeat sampling of the calibration area) and soil samples from the rest of the field were held out. Perhaps add a list that explains the three phases of the analysis (calibration, testing, and independent testing). Being clear about this will help emphasize the robustness of the calibration methodology.  

Line 173: Adding some background here as to why the Kennard–Stone algorithm was chosen and some brief background on latent variables and the role of hyperparameter tuning could be helpful for readers from an ecological background who are less familiar with these methodologies.

Table 2: Adding the statistics for the independent testing to this table could be useful to readers.

Figure 5: Adding the outlines of the calibration area and the sample points to these images could also be useful to readers.

Figure 5 and discussion: It could also be useful to include some information about the range of the soil attributes in the various groups of data (calibration, testing, and independent testing) utilized in this study. Perhaps that could explain some of the drop in performance when predicting secondary attributes.

Grammar:

Line 46: “based on low spatial resolution sampling grid” -> “based on a low spatial resolution sampling grid”

Line 49: “However, this is not is not feasible” -> “However, this is not feasible”

Line 94  “a 138-ha field at the municipality” ->  “a 138-ha field in the municipality”

Line 90: “and (ii) compare prediction” -> “and (ii) to compare the prediction”

Line 96: This sentence is very abrupt “Region of tropical climate with wet and dry season, on summer and winter, respectively”

Author Response

Thank you for carefully reading our study and for provide us insightful comments to improve the quality of the paper. The authors tried their best to improve to the original manuscript and fulfil the reviewer suggestions. Based on your comments, the manuscript was revised using the track changes function activated in Word.

Below you can follow our responses to each of your comments. 

  1. Comments: “Line 58: Defining on-line measurements could be useful as this article could be of interest to readers from different scientific backgrounds”

Answer: You are right. We have added an explanation about on-line measurements in L57, as you can also follow below:

“Alternatively, mobile platforms instrumented with proximal soil sensors acquire digital data about soil properties at fine scale through on-line measurements, i.e., with data acquisition performed on-the-go and at high frequency (e.g., one reading per second)”.

  1. Comments: “Lines 83-86: The main novel aspect of this paper is that the methods are being applied in Brazilian tropical soils rather than temperate regions. Some additional context discussing why this region is important and why these methodological developments are necessary could improve the introduction.”

Answer: Thank you for this comment. We have added some more information in the introduction (in L44-46 and L79-L87) to better characterize the research context.

  1. Comments: “Lines 168-170: I suggest editing this text and maybe also figure 1 for clarity to emphasize that the model was trained using the on-line surveys and 347 soil samples in the calibration area and that both the on-line survey data (which is a repeat sampling of the calibration area) and soil samples from the rest of the field were held out. Perhaps add a list that explains the three phases of the analysis (calibration, testing, and independent testing). Being clear about this will help emphasize the robustness of the calibration methodology.”

Answer: Thank you for your comment that allowed us to improve the explanation of the methodology used for calibration and validation of the models, fundamental for the understanding of the study as a whole. We have rethought about how to make this explanation clearer for the reader and decided to change different parts of the material and methods, please see L140-147, L194, L195-199 and L233-237. The main change was made in the Section 2.1., L140, as you can also follow below:

L140: “In summary, firstly the spectral data acquisition 1 was performed on the 18-ha area (designated as calibration area) followed by the collection of 347 soil in this same area (Figure 1D). This first data set (spectral data acquisition 1 + soil analysis of the 347 soil samples) was used for the calibration and testing of the predictive models (as detailed in Section 2.4.). Afterwards, the spectral data acquisition 2 was performed on the entire 138-ha field (including a resampling of the calibration area), followed by the collection of the 10 soil samples (Figure 1E). This second data set was used for independent testing of the previously calibrated models (as detailed in Section 2.5.), the data from the first data set was kept out of this new testing.”

  1. Comments: “Line 173: Adding some background here as to why the Kennard–Stone algorithm was chosen and some brief background on latent variables and the role of hyperparameter tuning could be helpful for readers from an ecological background who are less familiar with these methodologies.”

Answer: The Kennard–Stone algorithm was used to ensure that all attributes have comparable range and coefficient of variation for the sets used to calibrate and validate the models; in turn, this ensures that the observed prediction quality is related to sensor performance and not over- or underestimated due to diverging characteristics in the dataset. This information was better presented in L203-209, in the Results Section. We also added more information about the latent variables (L212-215) and hyperparameter tuning (L223-227). Regarding these parameters, a proper choice of them is very important in order to avoid underfitting (not all the data variance associated with the analytical task is taken into account) or overfitting (the data variance which is not associated with analytical task is taken into account).

  1. Comments: “Table 2: Adding the statistics for the independent testing to this table could be useful to readers.”

Answer: We agreed to add the complete statistics of the independent test. However, in order not to change the structure of Section 3.2. (where Table 2 is inserted), we have included this information in the Appendix Section (L472 and L478).

  1. Comments: Figure 5: Adding the outlines of the calibration area and the sample points to these images could also be useful to readers.”

Answer: We agree and added the outlines of the calibration area and the position of the sampling points to figure 5.

  1. Comments: Figure 5 and discussion: It could also be useful to include some information about the range of the soil attributes in the various groups of data (calibration, testing, and independent testing) utilized in this study. Perhaps that could explain some of the drop in performance when predicting secondary attributes.”

Answer: Thank you for this insight. We have added a figure (Figure A1 in L479) and a paragraph (L321) about the range of fertility attributes used in independent testing.

  1. Comments: Grammar:

Line 46: “based on low spatial resolution sampling grid” -> “based on a low spatial resolution sampling grid”

Line 49: “However, this is not is not feasible” -> “However, this is not feasible”

Line 94  “a 138-ha field at the municipality” ->  “a 138-ha field in the municipality”

Line 90: “and (ii) compare prediction” -> “and (ii) to compare the prediction”

Line 96: This sentence is very abrupt “Region of tropical climate with wet and dry season, on summer and winter, respectively””

Answer: Thank you. All recommended corrections have been applied to the text. We also revised the whole text, making improvements to the English

Finally, we would like to comment that we truly appreciate all your comments and we really made efforts to handle your concerns. We do hope the revised manuscript deserves your recommendation for Automation. 

If necessary, we continue disposed to make other alterations.

Sincerely,

The authors.

Round 2

Reviewer 1 Report

This article has been carefully revised and I have no further comments. 

Reviewer 2 Report

All of my comments have been well addresses and the text has been sufficiently revised.